# Inhibition of experimental autoimmune uveitis by intravitreal AAV-Equine-IL10 gene therapy

**Elizabeth Crabtree**[1], **Katy Uribe**[1], **Sara M. Smith**[1], **Darby Roberts**[1], **Jacklyn H. Salmon**[1],
**Jacquelyn J. Bower**[2,3], **Liujiang Song**[2,4], **Prabhakar Bastola**[2], **Matthew L. Hirsch**[2,4], **Brian C. Gilger**[1,2]*

1 Department of Clinical Sciences, North Carolina State University, Raleigh, North Carolina, United States of America, 2 Ophthalmology, University of North Carolina, Chapel Hill, North Carolina, United States of America, 3 Lineberger Comprehensive Cancer Center, University of North Carolina, Chapel Hill, North Carolina, United States of America, 4 Gene Therapy Center, University of North Carolina, Chapel Hill, North Carolina, United States of America

* bgilger@ncsu.edu

**Data Availability Statement:** All relevant data are within the paper and its Supporting Information files. Additionally, all supporting data and uncropped and unadjusted blot/gel images have

## Abstract

Equine recurrent uveitis (ERU) is a spontaneous, painful, and vision threatening disease affecting up to 25% of equine populations worldwide. Current treatments of ERU are non-specific and have many side effects which limits them to short-term use. In order to develop an effective therapy for ERU, we investigated the use of adeno-associated virus (AAV) gene therapy, exploiting a natural immune tolerance mechanism induced by equine interleukin-10 (Equine-IL10). The purpose of this study was to evaluate the therapeutic efficacy of a single intravitreal (IVT) dose of AAV8-Equine-IL10 gene therapy for inhibition of experimental auto-immune uveitis (EAU) in rats. Each rat was dosed intravitreally (IVT) in both eyes with either balanced salt solution (BSS) (control; n = 4), AAV8-Equine-IL10 at a low dose ($2.4 \times 10^9$ vg; n = 5) or high dose ($2.4 \times 10^{10}$ vg; n = 5). EAU was induced in all groups of rats 7 days after IVT injections and euthanized 21 days post-injection. Ophthalmic examination and aqueous humor (AH) cell counts were recorded with the observer blinded to the treatment groups. Histopathology and qPCR were performed on selected ocular tissues. Data presented herein demonstrate that AAV8-Equine-IL10 treated rats exhibited a significant decrease in clinical inflammatory scores and AH cell counts compared to BSS-treated EAU eyes on days 10, 12 and 14 post EAU induction at both administered vector doses. Mean cellular his-tologic infiltrative scores were also significantly less in AAV8-Equine-IL10 dosed rats com-pared to the BSS group. Intravitreal injection of AAV8-Equine-IL10 resulted in Equine-IL10 cDNA expression in the ciliary body, retina, cornea, and optic nerve in a dose-dependent manner. A single IVT injection of AAV8-Equine-IL10 appeared to be well-tolerated and inhibited EAU even at the lowest administered dose. These results demonstrate safety and efficacy of AAV8-Equine-IL10 to prevent EAU and support continued exploration of AAV gene therapy for the treatment of equine and perhaps human recurrent uveitis.

been uploaded to the Dryad database: Gilger, Brian (2022), AAV-eqIL-10 in EAU, Dryad, Dataset, https://doi.org/10.5061/dryad.q83bk3jkq.

**Funding:** BCG & MH: NC TraCS Grant (NC Translational and Clinical Sciences Institute) Grant number: # 550KR211915 https://tracs.unc.edu/index.php/services/pilot-program/tracs-5k-50k-grants The funders had no role in study design, data collection and analysis, decision to publish, or preparation of the manuscript.

**Competing interests:** The authors have declared that no competing interests exist.

## Introduction

Equine recurrent uveitis (ERU) is a spontaneous, painful and vision threatening non-infectious uveitis affecting up to 25% of equine populations worldwide [1]. This disease is characterized by episodes of active ocular inflammation alternating with varying intervals of clinical quiescence [2]. Research in horses, humans, and laboratory animals has shown that non-infectious uveitis is characterized by a T-cell mediated inflammatory response with a multifactorial origin, related to environmental factors and genetic makeup of an individual [3–5]. Recurrent uveitis in horses develops following primary uveitis when disruption to the blood-ocular barrier occurs, allowing CD4+ T-lymphocytes to enter and remain in the eye [6, 7]. This disruption enables host immune responses to react to ocular self-antigens that are not normally recognized by the immune system, and subsequent episodes of uveitis develop as a consequence of new antigenic detection [8]. The accumulated effects of recurrent 'bouts' or "flares" of inflammation leads to progressively destructive pathologic changes including irreversible scarring, ocular cloudiness, cataract formation, and vision loss [9]. Conventional treatment of ERU is non-specific, including frequent use of topical and systemic corticosteroids and other topical or oral immunosuppressive agents; none of which are effective in preventing uveitis relapses [1, 10]. These therapies are limited by poor treatment compliance and long-term adverse effects, such as corneal degeneration, glaucoma, cataract, ocular hypertension, and infection, all of which may contribute to the development of blindness [10, 11].

Experimental autoimmune uveitis (EAU) animal rodent models have been valuable in the study of non-infectious, immune mediated uveitis [12, 13]. EAU is mediated predominantly by CD4+ T- lymphocytes and proinflammatory cytokines, which are similar to the inflammatory profile in naturally occurring uveitis in humans and horses [7]. These models have proven useful in developing therapeutics to treat or prevent the immune inflammatory responses within the eye [14–16]. Blocking proinflammatory T-cells and enhancing anti-inflammatory T-cell cytokine function would halt the inflammatory cytokine cascade and therefore would be an effective target for treating immune mediated uveitis, such as ERU [15].

A immunomodulatory cytokine, interleukin-10 (IL10), has potent anti-inflammatory and immunomodulating properties that play critical roles in limiting the immune response and preventing autoimmune disorders [17–19] IL10 is produced by various immune cells and inhibits macrophages, natural killer, Th1, and dendritic cell function [20] IL10 also inhibits production of pro-inflammatory cytokines and co-stimulatory molecules, and interferes with general antigen presenting cell function [17, 20, 21]. Dysregulation of IL10 is associated with an increased response to infection and also an increased risk for development of many autoimmune diseases [17, 19, 22]. Several studies have evaluated the anti-inflammatory effects of IL10 both as a systemic and localized treatment [21, 23–29].

Gene therapy based on adeno-associated virus (AAV) has been clinically applied with optimistic results for multiple ocular diseases as a specific targeted long-term treatment [30–32]. AAV transduces many different cell types and establishes long-term transgene production following a single injection. Although there are several serotypes, AAV serotype 8 (AAV8) has been previously shown to effectively transduce ocular uveal tissue, an ideal location to target with gene therapy for the treatment of uveitis because it is the location of the blood ocular barrier [14, 33–35].

Current therapy for ERU requires long-term frequent application of topical medications, such as corticosteroids, which have poor compliance of use and frequent drug-related complications. Therefore, a single dose medication that mediates long-term immunosuppression specifically targeting the uveal tissue is desired. Here, we describe the use of an AAV vector encoding a codon optimized Equine-IL10 cDNA, a natural cytokine that is known to play a

major role in inducing ocular immune privilege, for the treatment of uveitis using AAV gene therapy. The purpose of this study was to evaluate the dose dependent therapeutic effects of gene therapy using AAV8-Equine-IL10 in a single intravitreal injection for the prevention of EAU in Lewis rats, a relevant and validated model of immune-mediated uveitis. Results of this study lay the groundwork for the development of an AAV-based gene therapy for long term treatment of patients, including horses, with non-infectious recurrent uveitis and support the use of AAV for the treatment of other inflammatory diseases of the anterior and posterior segment of the eye.

## Materials and methods

### Vector production, purification, and western blots

AAV8-Equine-IL10 was produced by the UNC Vector Core and characterized, as previously described [36]. The Equine-IL10 cDNA was codon optimized (GenScript USA, Piscataway, NJ). Western blot analysis was used to detect the size of the protein expressed; $5 \times 10^4$ cells per well were plated in a 24 well plate and incubated overnight in a humidified incubator at 37°C with 5% $CO^2$. Cells were transfected the next day with the Equine-IL10 plasmid using previously described PEI transfection protocol [37]. Forty-eight hours post transfection, supernatant was collected. 50 μl of supernatant was lysed with 50 μl of Mammalian Cell Lysis Buffer containing 1M Dithiothreitol (DTT) (2nM), 1M Tris (50nM), 100mM Ethylenediaminetetraacetic acid (5mM), 0.5% octylphenoxypolyethoxyethanol (0.0025%), 4M NaCl(100mM), Sigma Protease Inhibitor Cocktail (100uL), 1M beta-glycerolphosphate (10mM), 100mM Sodium Orthovanadate (1mM), 1M NaF (1mM), and distilled water. The supernatant was incubated for 30 mins at 4°C in a shaker. The supernatant solution was briefly sonicated (10 sec per sample) using the Branson Digital Sonifer®. Supernatant was collected following centrifugation at 10,000 g for 30 mins at 4°C, and cell lysates were prepared by mixing supernatant with 4X Laemmli Sample Buffer (BioRad#161–0747) + 10% 2-Mercaptoethanol (Sigma # M6250). The cell lysates were boiled at 85°C for 10 mins. Approximately 20 μg of total protein supernatant was loaded into each well of the SDS-PAGE polyacrylamide gel (BioRad#456–8083) and electroblotted into a Nitrocellulose membrane (BioRad#1620112) using the wet-transfer protocol. Electroblotted membrane was then blocked using 3% bovine serum albumin (Fisher#BP9703100) prepared in Tris-buffer saline with 1% Tween-20 (TBS-T) by shaking for 45 minutes at room temperature. The membrane was then incubated with Equine-IL10 primary antibody (R&D#AF1605) prepared in 3% BSA + TBS-T solution at a 1:1000 dilution and allowed to shake overnight at 4°C. Membrane was then washed with TBS-T solution thrice (5 minutes per wash) at room temperature, and subsequently was incubated with goat secondary IgG horseradish peroxidase (HRP)-conjugated secondary antibody (R&D#HAF017) prepared in 3% BSA (1:5000 dilution) in TBS-T for 1 hour at room temperature. Secondary antibody was aspirated, and the membrane was subsequently washed three times (5 minutes per wash) with TBS-T. Bands were visualized using the SuperSignal™ West Femto Maximum Sensitivity Substrate (Fisher, PI34096), and imaged using a GE AI600 Imager.

### Animals

The protocol for the use of animals was approved and monitored by the North Carolina University Institutional Animal Care and Use Committee (IACUC) and studies were performed according to the Association for Research in Vision and Ophthalmology Statement for the Use of Animals in Ophthalmic and Visual Research. Lewis rats (female, n = 14) (Charles River Labs, Wilmington, MA) were housed under 12/12-hour light/dark cycle in the NC State University Laboratory Animal Resources facility. Rats were randomly divided into 3 groups. Both

eyes were injected IVT with AAV8-Equine-IL10 at either a low dose of 2.4 x $10^9$ viral genomes (vg) (n = 5 rats; group 1), or a high dose of 2.4 x $10^{10}$vg (n = 5 rats; group 2) or balanced salt solution (BSS, Alcon Labs) (n = 4 rats) (see description of injections below). Experimental autoimmune uveitis (EAU) was induced in all rats 7 days following IVT injections and euthanized day 21 after IVT injections.

## Intravitreal administration

For IVT injections, rats were anesthetized with 2–3% Isoflurane (Henry Schein) in oxygen to effect. Topical anesthetic, proparacaine HCL 0.1% (Bausch and Lomb), was applied to each eye prior to IVT injection. Animals were placed in lateral recumbency (left eye injected first followed by the right eye). Each eye was cleaned with dilute 5% betadine solution. Intraocular injections were performed under an operating microscope using a Hamilton syringe (Hamilton Co.) and a 34-gauge stainless steel needle. Three microliters of either viral suspension (low dose or high dose) or BSS, mixed with 0.01% fluorescein sodium salt (Sigma) was administered IVT in both eyes, with the needle placement 1 mm posterior to the temporal limbus. After injections were completed, IVT injection was verified by presence of fluorescence in the vitreous followed by topical application of an antibiotic solution (Moxifloxacin 0.5%; Apotex Corp.) and ocular lubrication to the ocular surface to prevent infection and desiccation, respectively. Rats were allowed to recover from anesthesia on a heating pad until fully ambulatory.

## EAU induction and clinical evaluation of EAU

Seven days after IVT injection, rats were immunized subcutaneously at the base of the tail (100μg) and both thighs (50μg) with a 1:1 volume of human interphotoreceptor retinoid binding protein, IRBP (AnaSpec Inc., Fremont, CA) and Complete Freud's Adjuvant, CFA (Sigma-Aldrich, St. Louis, MO). Clinical assessment of ocular inflammation by slit lamp biomicroscopy (SL-17, KOWA) was performed daily (except for day 9 post IVT injection) with the examiner blinded to the treatment groups. Each eye was graded, according to previously described EAU scoring system: 0, normal; 0.5, dilated blood vessels in the iris; 1, abnormal pupil contraction; 2, hazy anterior chamber; 3, moderately opaque anterior chamber, but pupil still visible; 4, opaque anterior chamber and obscured pupil [13].

## Optical coherence tomography of the ocular anterior segment

Spectral domain optical coherence tomography (SD-OCT) was performed to image the anterior segment of each eye (Envisu R-class SD-OCT; Bioptigen, Inc, Morrisville, NC). The SD-OCT contains a super-luminescent light emitting diode delivering a wavelength of 840 nm. Imaging was performed using the hand-held probe of the SD-OCT device fitted with a noncontact 12-mm telecentric lens for image acquisition. After adjusting the arm reference length on the SD-OCT device by manufacturer recommendations, SD-OCT was set to 1000 A scans per B scan, and 100 B scans in total for each eye of each rat, to generate a radial volume of 4 mm in diameter. Each animal was manually restrained in right or left recumbency. Following imaging, cells in the anterior chamber were manually counted on 3 representative B scan images of each eye, as previously described [38]. Cell candidates within the anterior chamber were defined as at least two adjacent pixels with an intensity greater than a prespecified background threshold (S4 Fig) [38].

## Tissue collection

Animals were euthanized during peak disease activity (day 14 after induction of EAU) 21 days after IVT injections. Immediately after euthanasia, the left eye of each rat was dissected and tissues collected. A strict tissue collection and cleaning procedure were used between sample collections to minimize the potential of cross-contamination. Control rats (BSS injected) were dissected first followed by rats dosed with viral vector. Different sets of instruments were used to collect tissues for different treatment groups of rats. Instruments were cleaned with 70% alcohol, 5% sodium dodecyl sulfate detergent, and sterile saline between each sample taken. Upon collection, tissue samples were frozen on dry ice and then stored at −80˚C. Specific ocular sections of the left eye were collected included the cornea, conjunctival, iris/ ciliary body, retina, and optic nerve. The right eye was enucleated from each rat for histopathology. The right globes were fixed in 4% buffered paraformaldehyde overnight at 4˚C and then transferred to 70% ethanol before embedding in paraffin. Eyes were sectioned 5 μm through the optic nerve horizontal plane, and stained with hematoxylin and eosin. Infiltrative and structural grades of each eye were scored as previously described [39]. The globes were scored by two blinded observers separately and averaged to determine infiltrative and structural scores for each eye. Data are presented as mean +/− standard deviation (SD).

## Quantification of viral genomes by qPCR

DNA/RNA from the conjunctiva, cornea, retina, optic nerve, and ciliary body/iris were extracted with the AllPrep DNA/RNA Mini Kit according to the kit protocol (Qiagen, Valencia, CA). Vector biodistribution was quantitatively analyzed in the DNA fraction by qPCR utilizing UPL probe technology (MilliporeSigma). In short, the amount of vector- specific Equine-IL10 vector copies was standardized to a single copy of a rat GAPDH amplicon generated from total rat genomic DNA. For the detection of vector genomes, the plasmid (Equine-IL10) standard curve was generated by serial ten-fold dilutions in nuclease free water. qPCR was carried out with an initial denaturation step at 95˚C for 10 min, followed by 45 cycles of denaturation at 95˚C for 10 s, and annealing/extension at 65˚C for 30 s using Equine-IL10 specific primers (FWD: 5'-gaccagctggacaacatgc-3'; REV: 5'-actggatcatctcggacagg-3') and an internal fluorescent UPL probe #40 (5′- cttccca-3′). Genomic qPCR of rat GAPDH was performed with LightCycler® 480 SYBR Green Master Mix using the following primers: forward primer 5′-catttgatgttagcgggatct-3′; reverse primer 5′-tgggaagctggtcatcaac-3′. Serial dilutions of rat genomic DNA quantified via NanoDrop was used as a standard. The qPCR was carried out with an initial denaturation step at 95˚C for 10 min, followed by 45 cycles of denaturation at 95˚C for 10 s, and annealing at 57˚C for 30s. A melting curve analysis was performed at the end to ensure that a single product was amplified. Vector biodistribution data are reported as the number of vector Equine-IL10 copies per μg of gDNA.

## Quantification of Equine-IL10 expression by qPCR

Equine-IL10 gene expression was quantitatively analyzed by qPCR, similar to the vector genome methods described above. Briefly, RNA samples were treated with a DNA-free kit (Ambion) according to the manufacturer's instructions to remove any contaminating gDNA. First strand cDNA reverse transcription was performed using a High Capacity cDNA Reverse Transcription Kit according to the manufacturer's instructions (Applied Biosystems). For detection of Equine-IL10 transcripts, the same primers, probe, and cycling conditions were used as described above for the vector genome samples. qPCR of rat GAPDH cDNA was performed with LightCycler® 480 Probes Master Mix using the UPL Probe #60 and the same

primers and cycling conditions used for vector genomes as described above. Copies of Equine-IL10 transcript are reported as the number of copies per copy of host transcript (GAPDH).

## Statistical analysis

Comparisons of clinical and histologic scores from the experiments in this study were analyzed initially using the non-parametric Kruskal-Wallis test, i.e., to determine if at least one sample dominated. If there was a significant difference, then pairwise Wilcoxon (Mann-Whitney) tests were performed to evaluate for group differences using JMP version 14.0 (SAS Institute Cary, NC). To reduce between eye correlation, clinical and histologic scores and cell counts were averaged between eyes to provide a single clinical score or cell count per rat (and not per eye). qPCR data comparisons were generated using a one-way ANOVA and comparisons for each pair using a Student's t-test. Significance was set at $p < 0.05$ for all comparisons in this study.

## Results

### Equine-IL10 Western blot

Equine-IL10 cDNA was codon optimized and cloned into an AAV plasmid context. Western blots were used to demonstrate expression of Equine-IL10 in HEK293 cells (S1 Fig).

### AAV-Equine-IL10 gene therapy reduces inflammation in EAU rats

Rats were injected IVT in both eyes with either $2.4 \times 10^9$ viral genomes (vg) (n = 5 rats; group 1), $2.4 \times 10^{10}$ vg (n = 5 rats; group 2), or BSS (n = 4 rats; group 3) one week prior to induction of EAU. Clinical scores following IVT and prior to EAU induction revealed minimal ocular inflammation ($\leq 0.5$ clinical inflammatory score) that was not significantly different between groups. Following EAU induction, ocular inflammation in BSS-dosed rats developed on day 9 after IRBP injection and peaked on days 12 and 13 (Fig 1). Eyes dosed with IVT AAV8-Eq-IL10 consistently exhibited attenuated ocular inflammation as compared to the BSS dosed EAU rats (Fig 1). Mean EAU clinical scores were significantly lower in both the AAV Equine-IL10 low dose (days 10–14) and high dose (days 12–14) treated rats compared to the BSS rats (p = 0.002 to 0.049); Pairwise Wilcoxon tests). However, there were no significant differences in mean EAU scores between rats dosed with AAV Equine-IL10 high dose or low dose on any experimental day (Fig 1).

Optical coherence tomography (OCT) was performed prior to IVT injections, once prior to induction of EAU, and then every other day following induction of EAU (Fig 2). No AH cells were noted in any eyes prior to IVT injection or prior to EAU induction. Mean inflammatory OCT AH cell counts were significantly less in rats treated with IVT AAV8-Equine-IL10 (high or low dose) compared to BSS rats on days 10, 12 and 14 post EAU induction (p = <0.004 to 0.043); (Oneway ANOVA). There were no significant differences in mean inflammatory cell counts between AAV8-Equine-IL10 high or low dose treated rats on any experimental day.

Histological examination in BSS dosed EAU eyes demonstrated severe intraocular inflammation as evidenced by iris thickening, and severe inflammatory cell infiltration in the ciliary body, iris, aqueous humor and vitreous, as well as moderate vasculitis formation (Fig 3). However, in both the high and low dose AAV8-Equine-IL10 eyes, only a few scattered inflammatory cells were observed (Fig 3A). The mean infiltrative and structural histological scores were significantly less in the AAV8-Equine-IL10 low dose treated eyes (mean ±SD 1.0±0.0, 0.1±0.22 [infiltrative and structural, respectively]), as well as AAV8-Equine-IL10 high dose treated eyes

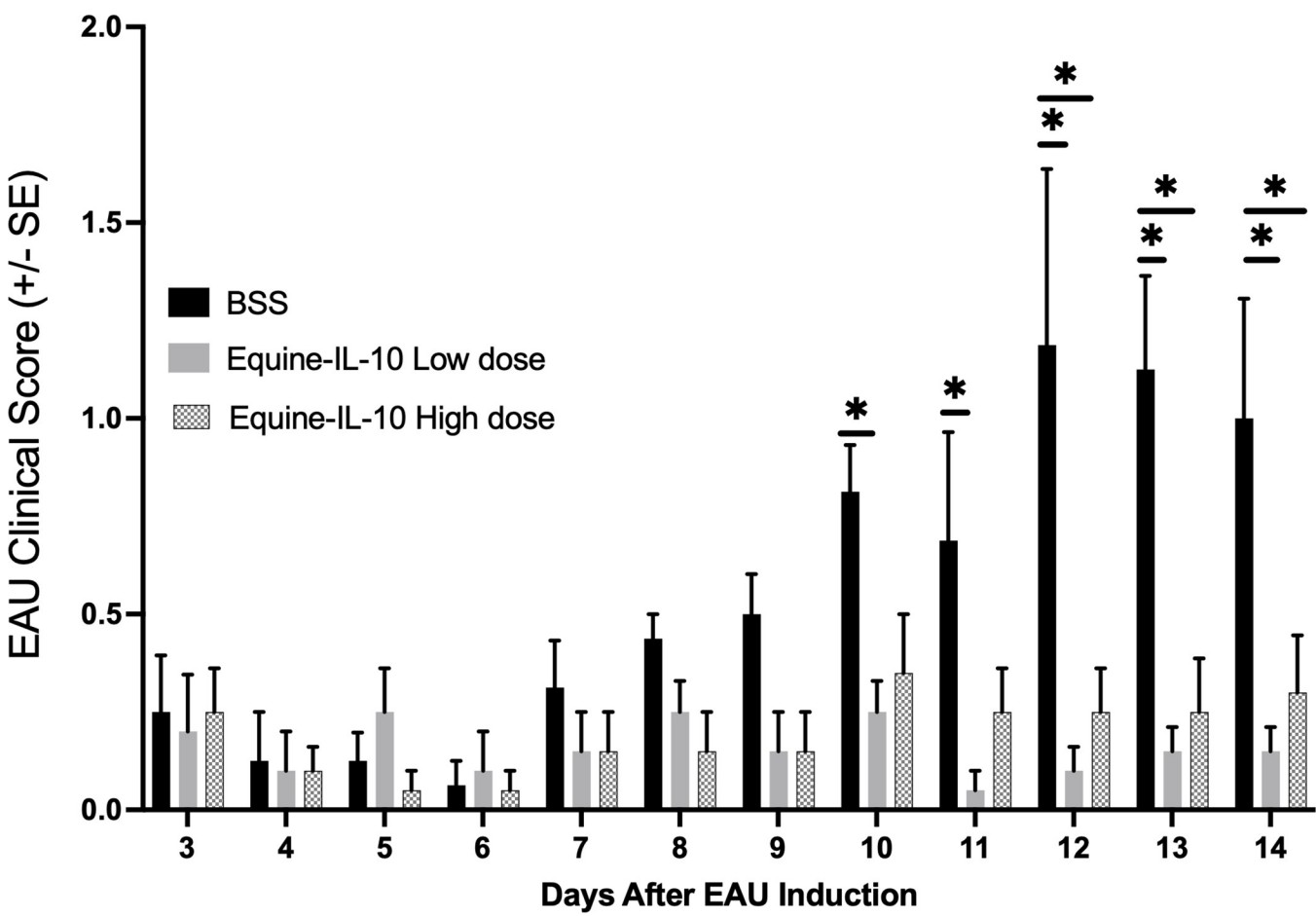

**Fig 1. AAV8-Equine IL10 Improves EAU clinical scores.** Rats were treated by intravitreal (IVT) injections with scAAV8-Equine-IL10 low dose or high dose (n = 5 rats, 10 eyes) or treated with BSS (n = 4 rats, 8 eyes). One week after IVT injections, EAU was induced by immunization with IRBP and ocular inflammation was examined by slit lamp biomicroscopy. (a) Bar graph of EAU mean clinical exam score for each rat (scores from each eye averaged to give one EAU score/rat) revealed that EAU clinical scores peaked on days 12, 13, and 14 after uveitis induction. Mean clinical scores were significantly lower on days 12–14 in the Equine-IL10 high dose (2.4x10^{10}vg) treated rats compared to the BSS rats and clinical scores were significantly lower on days 10–14 in the Equine-IL10 low dose (2.4x10^9vg) treated rats compared to BSS rats. (p = 0.002 to 0.049); Pairwise Wilcoxon tests). There were no significant differences in mean clinical scores between high dose or low dose rats treated with scAAV8-Equine-IL10 on any day.

(1.0±0.0, 0.1±0.22) compared to the non-treated BSS-EAU eyes (3.0± 1.15, 2.0±0.71) (Wilcoxon; p = 0.01, p = 0.015) (Fig 3B and 3C).

## Equine-IL10 ocular expression following intravitreal injection

Vector genome distribution of AAV8-Equine-IL10 in the conjunctiva, cornea, retina, optic nerve and iris/ciliary body was determined using qPCR from the left eye of each rat. Notably, in both low and high dose treated rats the highest vector copy number was found in the iris/ciliary body. In the low dose group (2.4 x10^9vg), vector genomes were recovered from the iris/ciliary body in 4/5 rats (80%), conjunctiva in 1/5 rats (20%), and retina in 1/5 rats (10%) (Fig 4B). The level of vector genomes were below the limit of detection for the optic nerve and cornea samples. In the high dose group (2.4x10^{10}vg), vector genomes were found in the iris/ciliary body 5/5 rats (100%), the conjunctiva in 4/5 rats (80%), the cornea in 4/5 rats (80%), the optic nerve in 3/5 rats (60%), and the retina in 5/5 rats (100%) (Fig 4B).

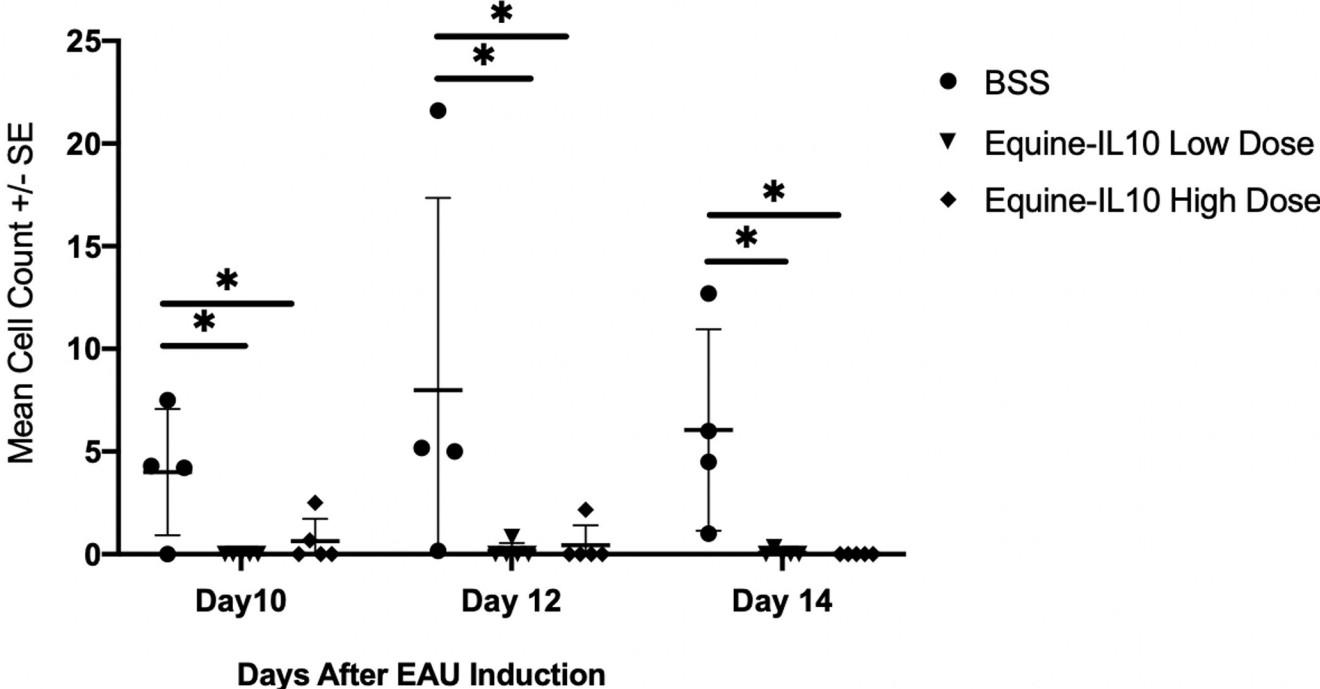

**Fig 2. AAV8-Equine-IL10 Improves EAU inflammatory cell count in the anterior chamber.** Optical coherence tomography (OCT) was performed once prior intravitreal injections, once prior induction of EAU and then every other day following induction of EAU. (a) scatter plot of EAU inflammatory cell count of each rat revealed that there was evidence of cellular infiltrate in the anterior chamber of rats on days 10, 12 and 14 following induction of EAU. Mean inflammatory cell count were significantly less in rats treated with a single intravitreal injection of scAAV8-Equine-IL10 (high or low dose) compared to BSS rats on days 10, 12 and 14 post EAU induction (p = <0.004 to 0.043); Pairwise Wilcoxon tests). There were no significant differences in mean inflammatory cell count between high or low dose treated rats on any day.

Equine-IL10 expression in treated rats was confirmed by qPCR using RNA isolated from the same ocular compartments. AAV8-Equine-IL10 high dose (2.4 x10$^{10}$ vg) group demonstrated significant equine-IL-10 expression in the iris/ciliary body, optic nerve, cornea, and retina compared to the low dose group and BSS controls (p<0.05) (Fig 4A). In the high dose group, Equine-IL10 transcript was detected in the iris/ciliary body 5/5 rats (100%), the cornea in 4/5 rats (80%), the optic nerve in 3/5 rats (60%), and the retina in 5/5 rats (100%) (Fig 4A). Although vector genomes were identified in 4/5 conjunctival samples in the high dose group, no equine-IL-10 cDNA was recovered in any of the conjunctival samples suggesting a low level of conjunctival expression.

There was no significant difference in equine-IL-10 expression between the low dose treated eyes and the control eyes in the conjunctiva, cornea, retina, optic nerve, and iris/ciliary body samples (Fig 4A). In the low dose group, though not significantly different compared to the control group, vector Equine-IL10 cDNA was detected in the iris/ciliary body 5/5 rats (100%), and the retina in 5/5 rats (100%) (Fig 4A). The level of equine-IL-10 cDNA transcripts from the conjunctiva, optic nerve, or cornea were below our limit of detection for the low dose group. Though the level of equine-IL-10 transcript was not significant in the ciliary body of the low dose group, clinically this group effectively reduced EAU clinical symptoms, suggesting that even very low levels of AAV-Equine-IL10 may suppress uveitis symptoms.

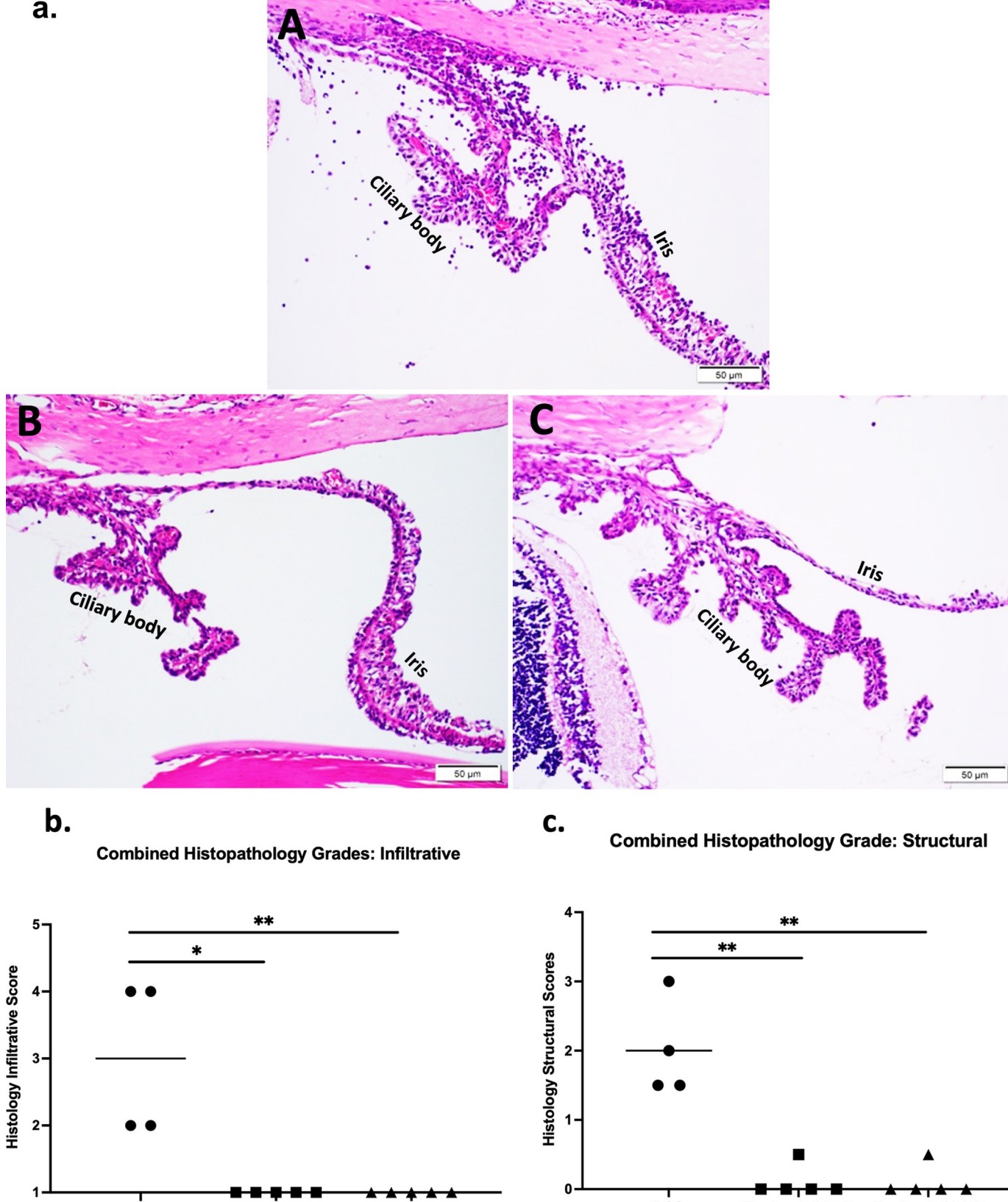

**Fig 3. AAV8-Equine-IL10 improves EAU histology scores.** (a) Representative images of ocular histology demonstrate iris thickening and severe inflammatory cell infiltration in the ciliary body, iris, anterior chamber and vitreous body, as well as moderate vasculitis formation in experimental autoimmune uveitis (EAU) BSS eyes (A). Mild to no infiltration of inflammatory cells was observed in the iris or ciliary body of high dose and low dose Equine-IL10 treated EAU eyes (B&C respectively). (hematoxylin & eosin staining). The histological infiltrative scores (b) and structural scores (c) were significantly decreased in both low and high dose AAV8-Equine-IL10 treated eyes (n = 5 eyes) as compared to the BSS- EAU eyes (n = 4 eyes) (p = 0.010, p = 0.015 respectively; Wilcoxon test). Each point is the average score of an individual eye of two blinded observers and mean scores of each treatment group are denoted by the horizontal bars. LD = low dose; HD = High dose.

In the control group, 1 conjunctival sample out of 20 ocular tissues sampled demonstrated vector genomes. The level of vector cDNA transcripts were below the limit of detection in the conjunctiva, cornea, optic nerve, retina, or iris/ciliary body in any of the control rats (Fig 4A & 4B). Thus, in the rats dosed with AAV-Equine-IL10 gene therapy, the vector was expressed in multiple ocular tissues suggesting that AAV-Equine-IL10 therapy correlated to the reduced EAU clinical and histologic scores observed in this study.

## Discussion

The results of this study demonstrate that AAV8- Equine-IL10, injected intravitrally in either a high (2.4 x$10^{10}$ vg) or low dose (2.4 x$10^9$ vg), suppressed the development of ocular inflammation in an EAU Lewis rat model. AAV8-Equine-IL10 IVT treatment resulted in vector Equine-IL10 cDNA detected in relevant ocular tissues, such as the iris/ciliary body and retina, which likely led to the reduction of clinical inflammatory scores, aqueous humor inflammatory cell counts, and histopathological scores observed in AAV8-Equine-IL10 treated versus saline treated rats. The results of the study reported herein support previous studies from our laboratories of the efficacy of AAV-mediated ocular gene transfer of immunosuppressive transgenes, such as human leukocyte antigen G (HLA-G), for the suppression of inflammation in the EAU rat model [14].

ERU is the leading cause of progressive blindness in horses with few effective and safe long-term preventative or treatment options [1, 10] Recurrent bouts of intraocular inflammation lead to progressive damage within the eye, which in turn, leads to significant economic loss and use for horse owners; as a consequence many affected horses are euthanized [1]. Herein, we discuss ocular gene therapy using AAV as a more effective treatment strategy for recurrent immune mediated ocular inflammation [3, 14]. This study supports that a single dose of AAV-mediated gene transfer of cDNA encoding an immunosuppressive cytokine, IL-10, may reduce or eliminate the need for uveitis treatment such as topical corticosteroids, systemic anti-inflammatories, and/or surgical interventions, and thus mitigate the risks of local or systemic immune suppression, while still reducing ocular inflammation and preserving long-term vision.

As demonstrated in this study and in previous studies, [14, 40] the eye has unique advantages for the use of AAV mediated gene addition. The eye is readily and easily accessible allowing for direct injection of the therapeutic. The eye also has tissue barriers that limit systemic redistribution of intraocular therapeutics [40]. In the present study, we demonstrate that both a high and lose dose IVT delivery of Equine-IL10 gene resulted in expression of Equine-IL10 cDNA in the ciliary body/iris and retina, which corresponded with decreased clinical and histological inflammatory scores in both treatment groups. Interestingly, the high dose group of rats also exhibited viral expression in the cornea and optic nerve, whereas the low dose groups did not. This indicates a dose dependent influence for viral distribution in ocular tissues via IVT injection.

Previous studies have demonstrated that several factors influence AAV's affinity for specific tissue transduction including different AAV serotypes and routes of injection [35] AAV8 has

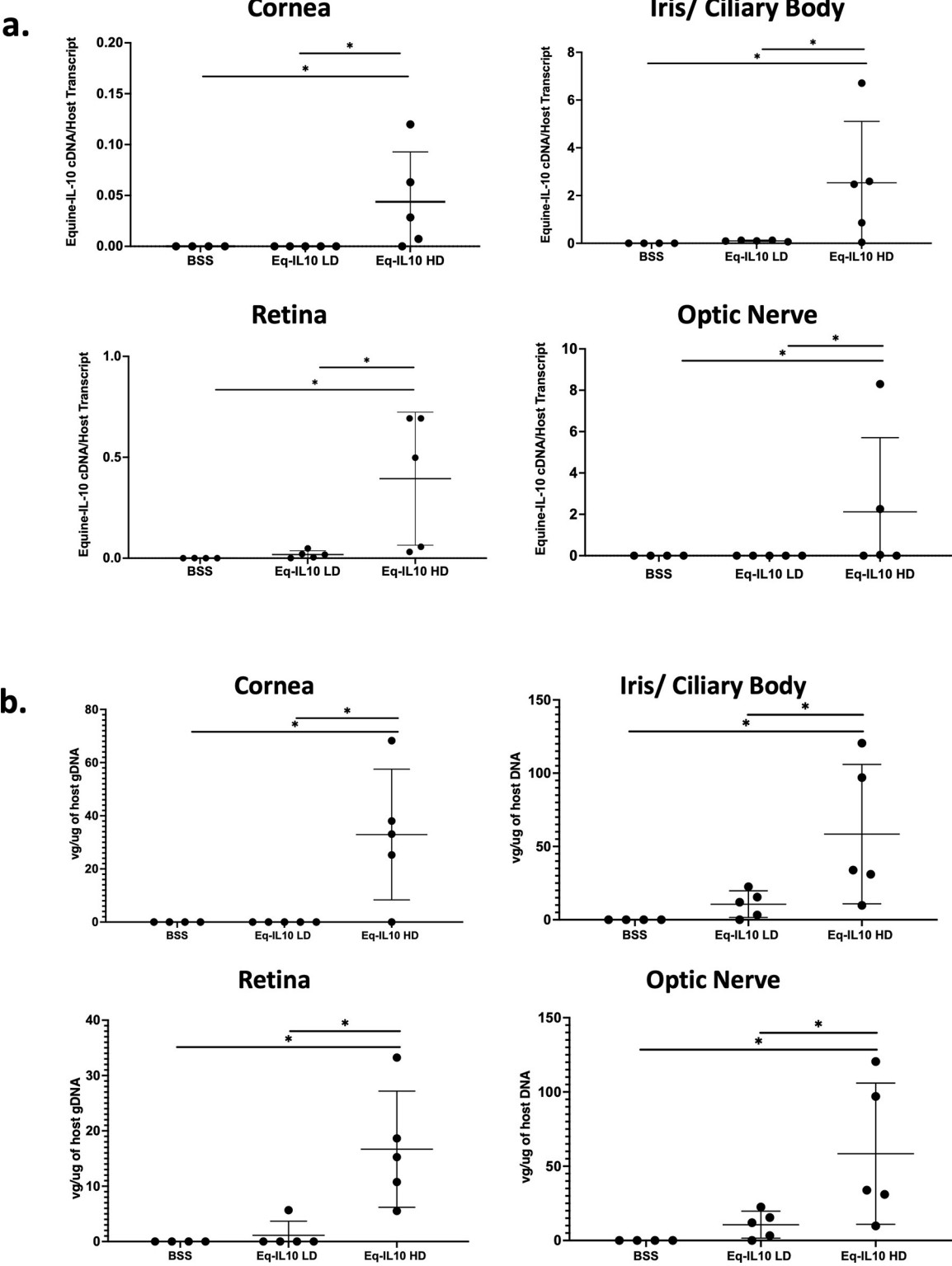

**Fig 4. AAV8-Equine-IL10 expression and ocular distribution.** (a) Equine-IL10 abundance examination by qRT-PCR in selected tissues are presented as vector cDNA /host transcript (GAPDH), One-way ANOVA ($^*p < 0.01$). Results represent experiments done in triplicate with mean value expressed. LD = low dose; HD = high dose; Conjunctival tissues were negative in all groups; (b) Vector genome copy number in distinct ocular tissues are shown as vector genome copy number/ug of host genome DNA, One-way ANOVA ($^*p < 0.01$).

an established tissue affinity for both the cornea and optic nerve following subconjunctival or intracameral injections [35] it is therefore not surprising that higher doses of viral genomes would also lead to increased frequency of off-targeted tissue transduction. It is important to note that the target tissue, the iris/ciliary body, had evidence of transduction in both the low and high dose groups in this study, though there was significantly more transduction in the high dose group compared to the low dose group. We target the ciliary body in this study because it is the location of the blood ocular barrier and a high concentration of localized IL10 at the blood ocular barrier theoretically would aid in stabilizing the barrier and maintain the eye's immune privileged state.

The IL10 immunoregulatory cytokine has been reported to function naturally as an important anti-inflammatory molecule. Dysregulation of IL10 in patients and rodent models is associated with an increased risk for development of many autoimmune diseases and infections [17, 19, 22]. Several studies have evaluated its anti-inflammatory effects as systemic and/or local treatment options [21, 23, 26–28]. Recent studies in horses, have already demonstrated promising results that intra-articular injection of gene therapy including AAV5-IL10 was effective in modulating synovial inflammation, with no systemic or localized adverse effects [23, 25, 41, 42].

Several studies have documented ranges of effective concentrations of IL10 to suppress T-lymphocytes [43–45]. According to Curto et al., 2016, horses diagnosed with uveitis and ERU had significantly higher concentrations of endogenous IL10 within their aqueous humor compared to normal horses [46]. When comparing monophasic uveitis to recurrent uveitis aqueous humor cytokine profiles in humans, intraocular IL10 increased during the remission of monophasic uveitis, and not in recurrent uveitis [47]. Based on these trends in cytokine profiles of uveitis, endogenous IL10 already plays an important role in ERU by protecting the eye from chronic inflammation and help preventing relapses of inflammation, thus suggesting that overexpression of IL10 may be promising therapeutic for ERU and other non-infectious uveitis. Subconjunctival injection of Adenovirus-IL10 effectively decreased the severity of experimental autoimmune uveoretinitis in mice [28]. Another study in 2008 demonstrated an intracameral injection of lentiviral-vector mediated expression of IL10 reduced inflammatory cell infiltrate and protein content in a mouse uveitis model suggesting that intraocular IL10 aids in maintaining integrity of the blood ocular barrier [29]. Gene therapy by adenoviral vectors appear to be short lived and incite local inflammation [48], and there is uncertainty for long term safety for lentiviral vectors [49]. This study therefore considered AAV as a more appropriate option for gene delivery in clinical patients. Other studies using AAV2 to deliver IL10 gene therapy to the eye for experimental uveitis have demonstrated success in reducing posterior uveitis models [27, 50]. The current study focused on EAU in the Lewis rat which is reliable and repeatable, established in our laboratory, [13, 14] and is representative of clinical uveitis. Despite the differences study designs, these studies further support that IL10 has an important role in the downregulation of ocular inflammation and in the maintenance of ocular immune privilege.

In the study reported herein, although effective in reducing ocular inflammation in EAU, viral transduction was at, or below, the detection limits of qPCR in ocular tissues recovered from the low dose treatment group. However, there was no significant differences of clinical exam scores or aqueous humor cell counts between the high and low dose group at any time point. This observation suggests that a low amount of Equine-IL10 is sufficient to reduce the severity of EAU and perhaps the earlier clinical response observed in the low dose group (compared to the high dose group) was related to a vector-associated inflammatory response in the high dose group reported to occur following intravitreal injection of high AAV vector doses [51]. Previous gene therapy dose escalation trials in humans have demonstrated potential

dose-dependent ocular inflammatory responses and increased neutralizing antibody titers following intravitreal or subretinal injections of doses over $1\times10^{11}$vg [52–54]. An alternative hypothesis is that there is limit to effective transduction of ocular tissue and the low dose AAV-IL-10 transduction was sufficient to locally inhibit the inflammatory response in EAU. This is an increasingly observed phenomenon in animal models and in human patients following intravitreal administration of AAV vectors [52–54]. Thus, a higher AAV concentration may not have resulted in higher IL-10 production in ocular tissues. Additional study is needed to determine the role of AAV vector dose and ocular inflammation. Furthermore, although the EAU model in Lewis rats is a well-established model for ERU in horses, the difference in globe size and volume between the two species is substantial. The specific dose and volume of viral vectors for an IVT injection into the horse globe is not well established and future studies are needed in order to establish an optimal anti-inflammatory dose of AAV-IL10 for larger eyes, such as equine or human eye.

Our interpretation of the data is that local immunosuppression in the eye induced by AAV gene transfer of IL-10 to ocular tissues prevented the inflammatory response characteristic of EAU. An alternative hypothesis is that the AAV-IL-10 gene therapy induced a generalized immunosuppression which, when administered prior to or during the immunization phase of EAU, may have prevented the development of an anti-IRBP T cell population and thus prevented EAU clinical signs. However, results from our previous studies of intravitreal administration of AAV gene therapy in rats demonstrated an overall lack of viral genome distribution to systemic tissues (14). These studies support that the amelioration of ocular inflammation in the EAU rat after AAV-IL10 (or HLA-G) is a local ocular immunosuppression effect. However, analysis of autoreactive T cells in IRBP immunized rats with and without AAV-IL10 IVT would confirm if the observed immunosuppressive effect on EAU is local or systemic, and should be done in further studies.

It is unknown whether long-term ocular Equine-IL10 expression following AAV gene transfer would cause any unwanted systemic or local adverse effect. However, equine-IL10 is already a natural component of ocular immune privilege in the horse [46], it is expected that this therapy of increased expression of Equine-IL10 would aid in restoring the natural ocular immune privilege state of the eye and not overtly increase risk of local infections; at least not more than the use of standard of care topical or systemic corticosteroids. The promising results described herein encourage further study to more thoroughly understand these initial findings.

The safety of AAV mediated gene therapy of ocular diseases has been investigated and established in human clinical trials [40]. More recent published reports suggest growing evidence that intraocular gene therapy can cause uveitis following IVT or subretinal administration [53, 55]. The design of the current study was such that intravitreal injections of gene therapy and BSS were performed 7 days prior to EAU induction. This was done so with the intent to discern a post injection inflammatory response in treated eyes as a reaction to the gene therapy or injection processes versus the intended induction of uveitis for this study purposes. Prior to EAU induction, there was no significant difference between all three groups on biomicroscopy clinical exam scores or OCT examinations. Also, as described above, rats treated with Equine-IL10 had lower clinical inflammatory scores and cellular counts on OCT images post EAU induction 7–21 days post IVT injections compared to control rats. It is possible that the immunosuppression from the Equine-IL10 could have suppressed any inflammation associated with the AAV intravitreal injections and decreased the T-cell response before and after EAU induction. Future toxicity studies of Equine-IL10 in healthy rat eyes would be necessary to confirm that the immunosuppressive transgene inhibits possible AAV induced inflammation.

The rat EAU model has a relatively acute inflammatory response, generally lasting 21 days. However, ERU uveitis and most other types of uveitis in humans is characterized as chronic or relapsing with durations of years [1, 9]. Models of chronic or relapsing uveitis exist and could be used to evaluate efficacy and safety of intravitreal gene therapy in future studies. Furthermore, our study design is a uveitis preventative model, while in clinical disease, most therapeutics would be given in established cases of uveitis. Further study is needed, either in clinical patients or chronic uveitis models to determine if similar treatment results using AAV-IL-10 will be observed.

In conclusion, this study demonstrates that IVT delivery of a single dose of AV8-Equine-IL10 established protection against ocular inflammation in EAU Lewis rats. Future clinical applications for AAV-Equine-IL10 in treating ERU is promising. Localized gene delivery of Equine-IL10 cDNA may significantly reduce the off-target risks associated with chronic use of systemic and topical anti-inflammatories. Intravitreal Equine-IL10 provides a long-term ocular anti-inflammatory therapeutic that would be an effective, novel therapeutic strategy for refractory and recurrent uveitis as well as other ocular autoimmune inflammatory diseases.

## Supporting information

**S1 Fig. Equine-IL10 Western blot.** A. A Western blot was used to detect Equine-IL10 protein following transfection of human embryonic kidney 293 cells (HEK293). Equine-IL10 protein (Equine-IL10) was detected in the supernatant of cultured HEK293 cells. KDa (kilodaltons). GFP (green fluorescent protein). p459 (Equine IL-10 plasmid). B. Uncropped raw image of Equine IL10 Western blot.
(ZIP)

**S2 Fig. Complete daily clinical exam inflammatory scores days 0–21.** Clinical examination inflammatory scores for daily examination after intravitreal injection. EAU was induced in all rats on 7 days after the intravitreal injections. Peak inflammation was seen on days 17–21 after intravitreal injections (10–14 days post EAU induction); * $p < 0.05$, Pairwise Wilcoxon tests.
(TIFF)

**S3 Fig. Representative images of retinal histopathology.** Representative images of ocular histology demonstrate inflammatory cell infiltration vitreous body, and retina in experimental autoimmune uveitis (EAU) BSS eyes, with almost infiltration of inflammatory cells observed high dose and low dose Equine-IL10 treated EAU eyes. (hematoxylin & eosin staining, original magnification: 20x).
(TIFF)

**S4 Fig. Representative OCT images of each group of rat on day 12 post EAU induction.** The red circles used to demonstrate cells in the anterior chamber. The iris, cornea and anterior chamber (AC) are labeled in the top right image.
(TIFF)

## Author Contributions

**Conceptualization:** Darby Roberts, Prabhakar Bastola, Matthew L. Hirsch, Brian C. Gilger.

**Data curation:** Elizabeth Crabtree, Katy Uribe, Sara M. Smith, Jacklyn H. Salmon, Jacquelyn J. Bower, Liujiang Song, Matthew L. Hirsch, Brian C. Gilger.

**Formal analysis:** Elizabeth Crabtree.

**Funding acquisition:** Matthew L. Hirsch, Brian C. Gilger.

**Investigation:** Elizabeth Crabtree, Matthew L. Hirsch, Brian C. Gilger.

**Methodology:** Elizabeth Crabtree, Matthew L. Hirsch, Brian C. Gilger.

**Resources:** Matthew L. Hirsch, Brian C. Gilger.

**Supervision:** Matthew L. Hirsch, Brian C. Gilger.

**Validation:** Jacquelyn J. Bower.

**Visualization:** Elizabeth Crabtree, Brian C. Gilger.

**Writing – original draft:** Elizabeth Crabtree.

**Writing – review & editing:** Elizabeth Crabtree, Jacquelyn J. Bower, Liujiang Song, Matthew L. Hirsch, Brian C. Gilger.

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
