## [Decision Letter · Decision Letter 0]

10 Mar 2022

PONE-D-21-37491Inhibition of experimental autoimmune uveitis by intravitreal AAV-Equine-IL10 gene therapyPLOS ONE

Dear Dr. Crabtree,

Thank you for submitting your manuscript to PLOS ONE. After careful consideration, we feel that it has merit but does not fully meet PLOS ONE’s publication criteria as it currently stands. Therefore, we invite you to submit a revised version of the manuscript that addresses the points raised during the review process. There are several critical issues raised by the reviews that include missing information in the methods and in the experimental approach. 

We look forward to receiving your revised manuscript.

Kind regards,

Andrew W Taylor, Ph.D.

Academic Editor

PLOS ONE

Journal Requirements:

5. Please upload a new copy of Figures 4 as the detail is not clear. Please follow the link for more information: " ext-link-type="uri" xlink:type="simple">https://blogs.plos.org/plos/2019/06/looking-good-tips-for-creating-your-plos-figures-graphics/"
https://blogs.plos.org/plos/2019/06/looking-good-tips-for-creating-your-plos-figures-graphics.

Reviewers' comments:

Reviewer's Responses to Questions

**Comments to the Author**

1. Is the manuscript technically sound, and do the data support the conclusions?

Reviewer #1: Partly

Reviewer #2: Partly

2. Has the statistical analysis been performed appropriately and rigorously? 

Reviewer #1: I Don't Know

Reviewer #2: Yes

3. Have the authors made all data underlying the findings in their manuscript fully available?

Reviewer #1: Yes

Reviewer #2: Yes

4. Is the manuscript presented in an intelligible fashion and written in standard English?

Reviewer #1: Yes

Reviewer #2: Yes

5. Review Comments to the Author

Reviewer #1: The authors address the significant unmet need for an effective, local, non-corticosteroid treatment option for non-infection uveitis. The goal of the study is to use a rat model of uveitis to study feasibility of their gene therapy approach prior to use in equine recurrent uveitis (ERU). The authors test the ability of two doses of an AAV8 vector expressing IL-10 to prevent EAU in Lewis rats. They conclude that intravitreal injection of both low and high dose vectors significantly decreased ocular inflammation and retinal damage. The manuscript is well written. The intro and discussion are thorough, and figures are appropriate. There are a few experimental design concerns that limit the significance of the conclusions. Specifically, that the treatment effect is local, and that the treatment approach is sufficient to treat uveitis in the presence of an established anti-retinal T cell population. Despite these concerns, this is an interesting study.

Experimental design:

1. Immunosuppression administered prior to or during the immunization phase of EAU can prevent development of the anti-ocular T cell population.

• In this study, it would be important to show that the local treatment approach (administered prior to uveitis induction) did not prevent the development of the expected autoreactive T cells. Ultimately for this gene therapy approach to be effective it must function in the context of the anti-ocular effector T cell population. Currently that data could also be interpreted to mean that the treatment prevented the generation of the anti-ocular T cell population at the time of immunization.

o Optional experiments would be to treat rats with IVT AAV8-IL-10 and then use adoptive transfer to generate EAU. Alternatively, T cell ELISPOT from immunized rats could be used to demonstrate the autoreactive T cells were present.

2. With the assumption that an autoreactive T cell population was generated as expected (see above concern), it is not clear that the beneficial effect of the gene therapy is local to the eye. The low dose treatment showed the same benefit as high dose treatment, but without similar evidence of local gene expression. An alternative hypothesis is that increased systemic IL-10 expression (in eye draining local lymph nodes or the spleen for example) could be responsible for the beneficial treatment effect. This could be addressed in the discussion and further mechanistic studies. Additional experiments to address this concern would be beyond the scope of this proof-of-concept study.

Statistical question:

• Was clinical score data (Figure 1) analyzed per eye or per animal? Please clarify. If per eye, there needs to be an adjustment to account for non-independence of eye level data, or score could be assigned per animal by averaging both eyes.

Comment:

Cell mediated killing of transduced cells is a concern for the safety and efficacy of gene therapy approaches. I was stuck by the appearance of the ciliary body in the high dose treatment group. There looks to have been considerable loss of CB tissue and thinning of the iris. Was this seen in multiple eyes, or is just this section abnormal?

Reviewer #2: This manuscript reports intravitreal administration of an IL-10-AAV8 vector 7 days prior to inducing EAU in Lewis Rats, and the authors were able to detect IL-10 transcripts in ocular tissues and found a reduction in EAU and in the inflammatory cell infiltration in the IL-10-AAV8-treated EAU rats relative to BSS-injected EAU controls. They conclude this to be an effective therapy which should be considered for treating equine recurrent uveitis.

Comments:

1. How many experimental replicates were performed in this study? It would appear from the text that only one experiment was carried out with 14 rats divided into three treatment groups. Due to the variability in EAU in rodent models, it is expected that any treatment effect be reproduced at least 3 times. Can the authors please clarify in the Results how many repeat experiments were carried out.

2. The use of the rat model in interesting as it is generally an acute form of ocular inflammation which eventually burns out over the course of a few weeks. Did the authors consider using a more recurrent form of EAU in mice? This might potentially be a more representative model with which to test this novel therapy.

3. Given that rat EAU is a very acute, and clinically severe form of uveitis, it is nevertheless impressive that a suppression of EAU was observed using both doses of vector. Did the authors consider administering the treatment 7 days AFTER inducing EAU? This is a commonly used approach to determine whether a treatment can down regulate an already established disease, and should be discussed in respect of the current study.

4. The histology images in Fig 3 are too faint in my version to easily detect the infiltrating cells, and should be taken at a higher resolution. The retinal layers appear relatively unaffected, but have become detached from the choroid. Is this an artefact of the fixation proceedure, or an effect of the EAU?

5. Whilst I agree that it is good to detect IL-10 in the ciliary body and iris, the blood-retinal barrier at the level of the retinal endothelial cells is also a key part of the barrier. It would be helpful if semi-quantitative data for the IL-10 PCR in each tissue was shown. Data in Fig 4 is difficult to understand, as there is no reference to IL-10, only vector cDNA. Whilst I appreciate that the safety issues were of huge concern, it is also important to know whether, at the larger concentration, IL-10 could in fact be detected within the retinal tissues.

6. PLOS authors have the option to publish the peer review history of their article (what does this mean?). If published, this will include your full peer review and any attached files.

Reviewer #1: No

Reviewer #2: No

---

## [Author Response · Author response to Decision Letter 0]

17 May 2022

Response to Reviewers, Manuscript: PONE-D-21-37491

Reviewer #1: The authors address the significant unmet need for an effective, local, non-corticosteroid treatment option for non-infection uveitis. The goal of the study is to use a rat model of uveitis to study feasibility of their gene therapy approach prior to use in equine recurrent uveitis (ERU). The authors test the ability of two doses of an AAV8 vector expressing IL-10 to prevent EAU in Lewis rats. They conclude that intravitreal injection of both low and high dose vectors significantly decreased ocular inflammation and retinal damage. The manuscript is well written. The intro and discussion are thorough, and figures are appropriate. There are a few experimental design concerns that limit the significance of the conclusions. Specifically, that the treatment effect is local, and that the treatment approach is sufficient to treat uveitis in the presence of an established anti-retinal T cell population. Despite these concerns, this is an interesting study.

Response: Thank you for your comments

Experimental design:

1. Immunosuppression administered prior to or during the immunization phase of EAU can prevent development of the anti-ocular T cell population. In this study, it would be important to show that the local treatment approach (administered prior to uveitis induction) did not prevent the development of the expected autoreactive T cells. Ultimately for this gene therapy approach to be effective it must function in the context of the anti-ocular effector T cell population. Currently that data could also be interpreted to mean that the treatment prevented the generation of the anti-ocular T cell population at the time of immunization. Optional experiments would be to treat rats with IVT AAV8-IL-10 and then use adoptive transfer to generate EAU. Alternatively, T cell ELISPOT from immunized rats could be used to demonstrate the autoreactive T cells were present.

Response: These are excellent comments and we added discussion points around this. Although unproven, our hypothesis is that the intravitreal IL-10 gene transfer and expression is limited primarily to the eye, which is supported by the overall lack of peripheral transgene or viral genome expression. 

2. With the assumption that an autoreactive T cell population was generated as expected (see above concern), it is not clear that the beneficial effect of the gene therapy is local to the eye. The low dose treatment showed the same benefit as high dose treatment, but without similar evidence of local gene expression. An alternative hypothesis is that increased systemic IL-10 expression (in eye draining local lymph nodes or the spleen for example) could be responsible for the beneficial treatment effect. This could be addressed in the discussion and further mechanistic studies. Additional experiments to address this concern would be beyond the scope of this proof-of-concept study.

Response: Also excellent comments and we added discussion points. Our hypothesis is that there is a limit to effective transduction of ocular tissue and the low dose AAV-IL10 was sufficient to locally inhibit the inflammatory response. In this case, more AAV is not better.

Statistical question:

Was clinical score data (Figure 1) analyzed per eye or per animal? Please clarify. If per eye, there needs to be an adjustment to account for non-independence of eye level data, or score could be assigned per animal by averaging both eyes.

Response: Thank you. We clarified the data analysis to indicated that scores were assigned to each individual rat, not eye, by providing an average score per animal. This eliminated the between-eye correlation. See changes in statistical analysis section and figure legends. 

Comment:

Cell mediated killing of transduced cells is a concern for the safety and efficacy of gene therapy approaches. I was stuck by the appearance of the ciliary body in the high dose treatment group. There looks to have been considerable loss of CB tissue and thinning of the iris. Was this seen in multiple eyes, or is just this section abnormal?

Response: We have re-done this histology figure with better representative images and labels. The changes observed in this section were interpreted to be processing and sectioning artifacts and not therapy related. A more representative image was provided. 

Reviewer #2: This manuscript reports intravitreal administration of an IL-10-AAV8 vector 7 days prior to inducing EAU in Lewis Rats, and the authors were able to detect IL-10 transcripts in ocular tissues and found a reduction in EAU and in the inflammatory cell infiltration in the IL-10-AAV8-treated EAU rats relative to BSS-injected EAU controls. They conclude this to be an effective therapy which should be considered for treating equine recurrent uveitis.

Comments:

1. How many experimental replicates were performed in this study? It would appear from the text that only one experiment was carried out with 14 rats divided into three treatment groups. Due to the variability in EAU in rodent models, it is expected that any treatment effect be reproduced at least 3 times. Can the authors please clarify in the Results how many repeat experiments were carried out.

Response: The rat EAU model is well established in our laboratory and results from these experiments are reasonably consistent (see PMID: 31882729). Based on these previous studies, we powered our study design to account for model induction variation. 

2. The use of the rat model in interesting as it is generally an acute form of ocular inflammation which eventually burns out over the course of a few weeks. Did the authors consider using a more recurrent form of EAU in mice? This might potentially be a more representative model with which to test this novel therapy.

Response: Yes, we did consider other models such as recurrent EAU models in mice. However, for initial proof of concept studies, we elected to use a larger animal, more consistent, model to evaluate response to intravitreal AAV therapy.

3. Given that rat EAU is a very acute, and clinically severe form of uveitis, it is nevertheless impressive that a suppression of EAU was observed using both doses of vector. Did the authors consider administering the treatment 7 days AFTER inducing EAU? This is a commonly used approach to determine whether a treatment can down regulate an already established disease, and should be discussed in respect of the current study.

Response: Yes, we did consider a treatment arm instead of a preventative arm. However, because it generally take 3-7 days for transgene expression to occur after injection of AAV, we felt that we may miss the therapeutic effect with the short duration of the EAU in this model. In the next phases of this study, use of the viral vector in a more chronic model after development of uveitis is planned. We added discussion points on this subject in the discussion. 

4. The histology images in Fig 3 are too faint in my version to easily detect the infiltrating cells, and should be taken at a higher resolution. The retinal layers appear relatively unaffected, but have become detached from the choroid. Is this an artefact of the fixation procedure, or an effect of the EAU?

Response: We have re-done this histology figure with better representative images and labels. The changes observed were interpreted to be processing and sectioning artifacts and not therapy or EAU related.

5. Whilst I agree that it is good to detect IL-10 in the ciliary body and iris, the blood-retinal barrier at the level of the retinal endothelial cells is also a key part of the barrier. It would be helpful if semi-quantitative data for the IL-10 PCR in each tissue was shown. Data in Fig 4 is difficult to understand, as there is no reference to IL-10, only vector cDNA. Whilst I appreciate that the safety issues were of huge concern, it is also important to know whether, at the larger concentration, IL-10 could in fact be detected within the retinal tissues.

Response: We apologize for the confusion – For these experiments, both RNA and DNA were isolated from the described ocular tissues. RNA was treated with DNAse and a first-strand cDNA synthesis kit was used to generate cDNA for each sample. Primers and a probe specific to the Equine-IL-10 sequence were used to amplify Equine-IL-10 transcripts using qPCR. Figure 4a depicts these results detecting transcripts specific to the equine form of IL-10 (i.e. vector cDNA in this case refers to the quantification of the equine IL-10 transcript isolated from each ocular compartment). The data presented in Figure 4b represents the qPCR results of the DNA isolated from each ocular tissue (both genomic DNA and vector DNA) using the same Equine-IL-10 specific primer/probe set and represents the biodistribution of the AAV-Equine-IL-10 vector.

Thus, as shown in Fig. 4a, the Equine-IL-10 transcript was detected in the retinal tissue at the high dose. We have amended the text in the results section and modified the figure legends to enhance the clarity of these results.

While revising your submission, please upload your figure files to the Preflight Analysis and Conversion Engine (PACE) digital diagnostic tool, https://pacev2.apexcovantage.com/. PACE helps ensure that figures meet PLOS requirements. 

Response: We uploaded and formatted our figure files to the PACE digital diagnostic tool. The submitted figures were those generated by PACE.

---

## [Decision Letter · Decision Letter 1]

22 Jun 2022

Inhibition of experimental autoimmune uveitis by intravitreal AAV-Equine-IL10 gene therapy

PONE-D-21-37491R1

Dear Dr. Gilger,

We’re pleased to inform you that your manuscript has been judged scientifically suitable for publication and will be formally accepted for publication once it meets all outstanding technical requirements.

Kind regards,

Andrew W Taylor, Ph.D.

Academic Editor

PLOS ONE

Additional Editor Comments (optional):

Reviewers' comments:

Reviewer's Responses to Questions

**Comments to the Author**

1. If the authors have adequately addressed your comments raised in a previous round of review and you feel that this manuscript is now acceptable for publication, you may indicate that here to bypass the “Comments to the Author” section, enter your conflict of interest statement in the “Confidential to Editor” section, and submit your "Accept" recommendation.

Reviewer #1: All comments have been addressed

2. Is the manuscript technically sound, and do the data support the conclusions?

Reviewer #1: Yes

3. Has the statistical analysis been performed appropriately and rigorously? 

Reviewer #1: Yes

4. Have the authors made all data underlying the findings in their manuscript fully available?

Reviewer #1: Yes

5. Is the manuscript presented in an intelligible fashion and written in standard English?

Reviewer #1: Yes

6. Review Comments to the Author

Reviewer #1: The authors have been very responsive to reviews. My concerns have been addressed. Statistical concerns have been clarified and new histology images addressed my concerns about a toxic effect of treatment.

7. PLOS authors have the option to publish the peer review history of their article (what does this mean?). If published, this will include your full peer review and any attached files.

Reviewer #1: **Yes: **Kathryn Pepple, MD, PhD

---

## [Editor Report · Acceptance letter]

2 Aug 2022

PONE-D-21-37491R1 

Inhibition of experimental autoimmune uveitis by intravitreal AAV-Equine-IL10 gene therapy 

Dear Dr. Gilger:

I'm pleased to inform you that your manuscript has been deemed suitable for publication in PLOS ONE. Congratulations! Your manuscript is now with our production department. 

Kind regards, 

on behalf of

Dr. Andrew W Taylor 

Academic Editor

PLOS ONE